

SciPost Phys. Lect. Notes 76 (2023)

# Plea for the use of the exact Stirling formula in statistical mechanics

Didier Lairez⋆

Laboratoire des solides irradiés, École polytechnique,
CEA, CNRS, IPP, 91128 Palaiseau, France

⋆ didier.lairez@polytechnique.edu

## Abstract

In statistical mechanics, the generally called Stirling approximation is actually an approximation of Stirling's formula. In this article, it is shown that the term that is dropped is in fact the one that takes fluctuations into account. The use of the Stirling's exact formula forces us to reintroduce them into the already proposed solutions of well-know puzzles such as the extensivity paradox or the Gibbs' paradox of joining two volumes of identical gas. This amendment clearly results in a gain in consistency and rigor of these solutions.

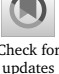

## 1 Introduction

Classical and phenomenological thermodynamics is only concerned with equilibrium : *"A system is in an equilibrium state if its properties are consistently described by thermodynamic theory"* (H. B. Callen [1] p. 15). This must be understood *stricto sensu*. For example, defining equilibrium as the state that maximizes thermodynamic (Clausius) entropy does not make sense because the value of entropy at equilibrium cannot be compared to those close to equilibrium, as there is no way to calculate the latter. Classical thermodynamics does not consider fluctuations. It is a major contribution of statistical mechanics to have introduced this notion *via* probabilities. For this reason it is quite intriguing that a rough approximation of the Stirling formula for the asymptotic behavior of $\ln(n!)$ is commonly used, namely the Stirling approximation : $\ln(n!) \simeq n \ln(n/e)$. Because we will see that this approximation forces us precisely to neglect the fluctuations.

The purpose of this article is to demonstrate this inconsistency, through two examples of conflicts or paradoxes between statistical mechanics and thermodynamics : 1) that of extensivity of entropy; 2) one of Gibbs' well-known paradoxes that concerns the joining of two identical gases. The latter is very often expressed in terms of the former, but I believe that considering them independently increases their heuristic value. We will see that the usual treatments of these paradoxes benefit from the exact cancellation of two errors : 1) neglecting fluctuations in the statement of the problem; 2) using a rough approximation of the Stirling formula. Thus, from a logical point of view, these paradoxes are not resolved with these usual treatments, but can be if we properly reconsider the two above points.

## 2 Stirling formula and its approximation

In statistical mechanics, we are often confronted with the calculation of $n!$ in the thermodynamic limit $n \to \infty$, or in terms of its logarithm :

$$\ln(n!) = \ln(1 \times 2 \times 3 \cdots \times n) = \sum_{k=1}^{n} \ln(k). \tag{1}$$

A first rough approach to compute this sum would be to approximate it by an integral :

$$\ln(n!) \simeq \int_0^n \ln(x)\,\mathrm{d}x = n \ln(n/e). \tag{2}$$

But a better one is based on the idea that an integral is squeezed between two Rienmann sums, or equivalently that the sum is squeezed between two integrals. This leads to :

$$\begin{aligned} \ln(n!) \;&\simeq \int_0^n \ln(x)\,\mathrm{d}x + \frac{1}{2}\ln(n) \\ &\simeq n \ln(n/e) + \frac{1}{2}\ln(n). \end{aligned} \tag{3}$$

The factor $\frac{1}{2}$ can be viewed as a variation of the middle-point rule. In fact, the sequence $\ln(n!) - n \ln(n/e)$ is not convergent, contrary to $u_n = \ln(n!) - [n \ln(n/e) + \frac{1}{2}\ln(n)]$. The actual calculation of the latter's limit is attributed to Stirling, who proved that $\lim_{n \to \infty} u_n = \frac{1}{2}\ln(2\pi)$ (see [2] p. 52). So that we can finally write :

$$\ln(n!) = n \ln(n/e) + \ln(\sqrt{2\pi n}) + o(1). \tag{4}$$

This is the exact Stirling formula whereas Eq. 2 will be called Stirling approximation in the following.

For the purpose of this paper, the interesting point is that a discerning reader can recognize in the second term of Eq. 4 the Shannon differential entropy of a Gaussian distribution with standard deviation $\sqrt{n}$ (for a derivation see [3] p. 243). The usual way to demonstrate the Stirling's result passes *via* Wallis' integrals (see [4] p. 616), but actually the exact Stirling formula can also be derived from the central limit theorem [5] introducing quite naturally fluctuations.

The use of the Stirling approximation (Eq. 2) is usually justified by the fact that $\ln(\sqrt{2\pi n})$ is negligible compared to $n\ln(n/e)$ in the thermodynamic limit (see e.g. [6] p. 497, [7] sec.3.3.2). True, but in many cases $\ln(\sqrt{2\pi n})$ should be compared to zero instead.

# 3 The paradox of extensivity

## 3.1 Extensivity in thermodynamics

Let us consider entropy as a function $\mathcal{S}$ of the three variables $U$ the internal energy, $V$ the volume and $N$ the number of particles. $\mathcal{S}$ is extensive if

$$\forall \alpha \in \mathbb{R}, \quad \mathcal{S}(\alpha U, \alpha V, \alpha N) = \alpha \mathcal{S}(U, V, N). \tag{5}$$

For $\alpha = 1/N$, one obtains the equality

$$\mathcal{S}(U, V, N) = N\mathcal{S}(U/N, V/N, 1). \tag{6}$$

So that, by defining $u = U/N$ and $v = V/N$ as the internal energy and the volume per particle, respectively, one can write :

$$\mathcal{S}(U, V, N) = Ns, \quad \text{where} \quad s = \mathcal{S}(u, v, 1). \tag{7}$$

If $u$ and $v$ are constant, then $s$ is too and $\mathcal{S}(N)$ is the sum of $N$ individual constant contributions. In short, a quantity is extensive if it varies proportionally to $N$ while intensive quantities (e.g. density $N/V$) are held constant.

Now envisage a system, say a simple gas, made of two disjoined subparts $A$ and $B$ with additive state-variables such as

$$\begin{aligned} U_A &= \alpha U, & V_A &= \alpha V, & N_A &= \alpha N, \\ U_B &= (1-\alpha)U, & V_B &= (1-\alpha)V, & N_B &= (1-\alpha)N. \end{aligned}$$

As entropy is additive, the total entropy $S_{A\cup B}$ of the mathematical union (see Fig.1) of these disjoined subparts is :

$$S_{A\cup B} = \mathcal{S}(U_A, V_A, N_A) + \mathcal{S}(U_B, V_B, N_B). \tag{8}$$

In addition, if entropy is extensive, using Eq. 5 one has

$$\begin{aligned} \mathcal{S}(U_A, V_A, N_A) &= \alpha \mathcal{S}(U, V, N), \\ \mathcal{S}(U_B, V_B, N_B) &= (1-\alpha)\mathcal{S}(U, V, N), \end{aligned}$$

leading to

$$\mathcal{S}(U, V, N) = \mathcal{S}(U_A, V_A, N_A) + \mathcal{S}(U_B, V_B, N_B). \tag{9}$$

$\mathcal{S}(U, V, N)$ is the entropy $S_{J(A,B)}$ of the system that would be obtained by the physical joining of $A$ and $B$, that is without physical separation between them (see Fig.1). Finally, Eq. 8 and 9 give :

$$S_{J(A,B)} = S_{A\cup B}. \tag{10}$$

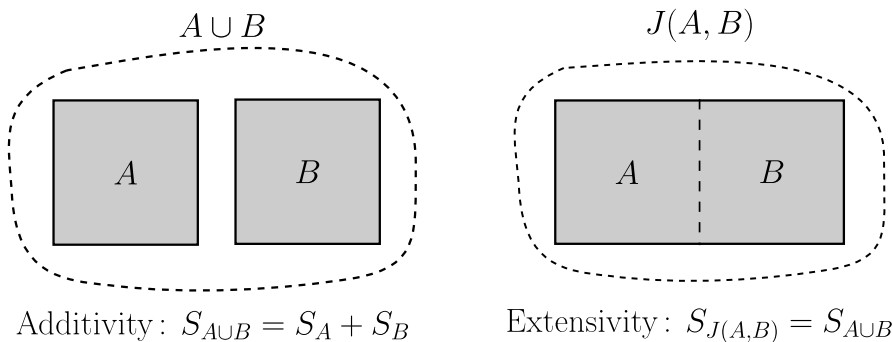

Figure 1: Mathematical union versus physical joining.

In thermodynamics, it is quite usual to write the fundamental equality as $dS = \frac{1}{T}dU + \frac{P}{T}dV - \frac{\mu}{T}dN$, where $T$ is the temperature, $P$ the pressure and $\mu$ the chemical potential. This equality would suggest that in thermodynamics $S$ is a function of $N$. Actually it is not, because $\mu$ is an unknown function of $N$. It is the contribution of statistical mechanics [8] to have made $\mu$ an explicit function of $N$. Therefore, extensivity of entropy should not be a relevant question in thermodynamics. Actually, there is no experimental evidence for extensivity, i.e. there is no reversible transformation that would allow $S$ to be measured while $N$ would vary (this had already been mentioned by E. Einstein in 1916 [9] p. 125). Nor is there evidence to the contrary. Extensivity of entropy is an undecidable question in phenomenological thermodynamics. That is to say, whether we consider it to be or not has no bearing on the way in which thermodynamics can account for phenomena.

That being said, Eq. 7 is a very interesting property allowing great mathematical simplifications of many problems of thermodynamics (via Euler and Gibbs-Duhem equations) [1] in the thermodynamic limit of very large $N$ and $V$ when surface (finite size) effects can be neglected. But above all, Eq. 10 is considered essential to build an axiomatic thermodynamics [1, 6, 10]. Inspired by how theories are constructed in other areas of physics (especially statistical mechanics), the aim of this approach is to distance thermodynamics from its initial phenomenological approach (i.e. derive laws from experiments) and to rebuild everything from initial postulates (i.e. put the laws first). One of the main postulates is that the entropy of a system is maximized at equilibrium. This idea is very natural, imprinted as we are on our experience of mechanics. It is also a result of statistical mechanics. In thermodynamics, conjointly with the other postulate that entropy cannot spontaneously decrease (traditionally the second law of thermodynamics), the postulate that entropy is maximized at equilibrium has two advantages: 1) it ensures the stability of the equilibrium [1] (without having to postulate it); 2) it makes it possible in principle to derive the equilibrium conditions by maximizing the entropy. The problem is that maximizing the entropy of a system assumes that we are able to compute it under non-equilibrium conditions. The equations of thermodynamics do not allow this, unless entropy is assumed to be extensive. Consider the two previous disjoined subparts $A$ and $B$. They are necessarily in equilibrium with each other because they are independent. So that the entropy of the disjoined system can be computed regardless of the values of $(U_A, V_A, N_A)$ and $(U_B, V_B, N_B)$. By using extensivity and Eq. 10 the entropy of the joined system is deduced, even if the values for $(U_A, V_A, N_A)$ and $(U_B, V_B, N_B)$ do not correspond to an equilibrium condition of the joined system (for examples of such calculation see e.g. [1] chap. 2).

This is the main reason why, in the thermodynamic limit, extensivity is postulated in axiomatic thermodynamics.

## 3.2 The paradox and its usually proposed solutions

In statistical mechanics, entropy is written as an explicit function of $N$, which makes it possible to check if it is extensive, in accordance with the postulate of axiomatic thermodynamics.

Consider a system made of $N$ particles of gas in a closed volume $V$ expressed in unit of $\lambda^3$, with $\lambda$ the thermal length of de Broglie. Let us express the temperature $T$ in Joule, so that the entropy has no dimension. The Boltzmann entropy is:

$$S = \ln V^N = N \ln V. \tag{11}$$

Thus, if $N$ increases at a constant density $V/N$, $V$ must also increase. It follows that the Boltzmann entropy is not proportional to $N$ and is not extensive. Hence the paradox or conflict with axiomatic thermodynamics (not with phenomenological thermodynamics).

Before examining how the paradox is usually solved, note that: 1) axiomatic thermodynamics was built to match with statistical mechanics, in particular with the postulate of maximum entropy at equilibrium; 2) this imposes the postulate of extensivity; 3) but statistical entropy is not always extensive; 4) so that the ball is in the court of statistical mechanics which is urged to reconsider its calculation in order to match thermodynamics. This all sounds like fallacious circular reasoning which should be enough to disregard the paradox. But a paradox, by the simple fact that it can be stated, offers the opportunity to deepen the theory.

In my knowledge all solutions to the extensivity paradox amount to finally write instead of Eq. 11:

$$S = \ln \frac{V^N}{N!}, \tag{12}$$

that is called the correct Boltzmann counting. So that, by using the Stirling approximation (Eq. 2) one obtains:

$$S = N \ln \frac{Ve}{N} = Ns, \tag{13}$$

with

$$s = \ln \frac{Ve}{N}, \tag{14}$$

is constant. These last two equations express the extensivity of the entropy that we were looking for.

Justifications for Eq. 12 will be discussed in the next section. Here, let us just use the exact Stirling formula (Eq. 4) instead of its approximation (Eq. 2), Eq. 12 transforms into:

$$S = N \ln \frac{Ve}{N} - \ln(\sqrt{2\pi N}) = Ns - \ln(\sqrt{2\pi N}), \tag{15}$$

instead of Eq. 13. The difference $S - Ns$ diverges and cannot be neglected as it should be actually compared to zero. In other words, $Ns$ is actually not an asymptote of $S$ and Eq. 12 does not make entropy extensive in the thermodynamic limit.

It is quite strange to observe that the Stirling exact formula is well known and quoted in many textbooks of statistical mechanics (e.g. [6, 7]), but that in the same texbooks the idea that $\lim_{N\to\infty}[S - Ns]$ is finite still persists ( [7] sec. 4.2), as well as the idea that the correct Boltzmann counting makes entropy extensive ( [6] p. 268 and 497). Sekerka goes even further and justifies the use of Stirling's approximation like this. "*Other terms* [i.e. other than $N \ln(N/e)$] *in Stirling's approximation* [exact Stirling formula in this paper] *have been dropped because they would lead to sub-extensive results*" [6] p. 261. This is clearly a circular reasoning.

### 3.3 Amendment to the solution

The correction of the counting of microstates by $N!$ was first introduced by Gibbs (see [11] chap. XV) but the justification was quite obscure. Gibbs was aware that dividing the number of possible microstates by $N!$ amounts to considering that particles are exchangeable and lose their individuality, which was not easily conceivable in his time. Then the ideas of quantum mechanics emerged which made it "conceivable" that, under certain circumstances, particles were inherently and conceptually indistinguishable. So the idea was accepted that the term $-\ln(N!)$ in the entropy cannot be understood in a classical manner but has a quantum origin (see e.g. [12] p. 141, [10] p. 115). In fact, it has long been shown that $-\ln(N!)$ can also be obtained in the classical framework [13–15]. As everything from the beginning in statistical mechanics has been built "classically", for self-consistency this latter solution is the best for classical particles, which are always distinguishable, in the sense that there is no conceptual impossibility to follow their trajectories, and then to preserve their individuality or identity.

The essence of the classical-framework derivation of the correct Boltzmann counting is as follows. Isolated systems (microcanonical ensemble with Boltzmann entropy) and closed systems (canonical ensemble with Gibbs entropy) have constant number of particles $N$. Thus, any multiplicative factor applied to the number of microstates that should be a function of $N$ (such as $1/N!$) would finally results in an additional constant in the entropy. This constant will vanish when calculating the change in entropy of the system when it undergoes any process (remember that thermodynamical experiments give only access to entropy differences). Actually, varying $N$, by applying a scaling factor as in Eq. 5, only makes sense for an open system (grandcanonical ensemble). It has been shown that computing entropy in the grandcanonical ensemble gives the correct Boltzmann counting [13–17].

Interestingly, one can wonder how it is possible that distinguishable particles in an open system give the same result as indistinguishable particles in an isolated system? The answer is quite simple. In an open system, particles are continuously renewed by exchanges with the surroundings. In other words, individual particles at a given time are not the same as those a little later. The fact that we consider an open volume of gas as a "system" implies that it is totally independent of the identity of the particles of which it is made up and that its integrity is not affected by the exchange of certain particles by others of the same species. The identity of particles is an unnecessary information for its description. This opens the door to an interpretation of the correct Boltzmann counting precisely in terms of information.

Let us consider the state of the system as a random variable and its entropy as a measure of uncertainty about its outcome [18–20]. Uncertainty or missing information about this outcome. At a microscopic level, states are microstates and the outcome that the system is currently adopting. The greater the number of microstates, the greater the uncertainty. This is the meaning of the Boltzmann entropy in an isolated system (i.e. a special case of statistical (Gibbs' or Shannon's) entropy of uniform distribution). In this case, a given microstate is described (identified) by the momentum and position of each particles. If we do not care about the identity of particles because the system is open, which specific microstate the system is currently adopting among the $N!$ possible permutations of particles is not a relevant information. This is the meaning of the quantity $\ln(N!)$ that must be subtracted to the total information that we are missing. But there is another source of uncertainty, independent of the previous one, that of the fluctuating number $N$ of particles of which the system is made up. This number obeys a binomial distribution, which (according to the central limit theorem) tends toward a Gaussian distribution of standard deviation $\sqrt{N}$ ([21] p.9) and entropy $\sqrt{2\pi N}$. Finally the uncertainty (the entropy) is:

$$S = N \ln V - \ln(N!) + \ln(\sqrt{2\pi N}). \qquad (16)$$

So that by using the exact Stirling formula (Eq. 4) we obtain Eq. 13 but following a way which

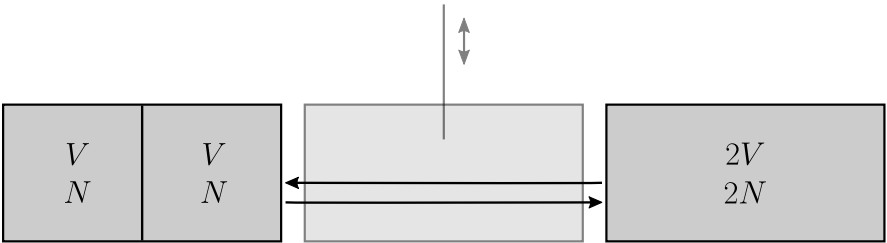

Figure 2: Joining/disjoining cycle of two volumes of the same gas at the same temperature and the same pressure as viewed in thermodynamics. The joining process produces no difference in Clausius entropy.

is more rigorous ($Ns$ is now an asymptote of $S$) and consistent (an open system cannot be conceived without fluctuations).

Incidentally, this solution of the extensivity puzzle emphasizes that extensivity in only obtained for open fluctuating systems, consistently with its definition as a scaling property.

## 4 The Gibbs paradox on joining two volumes of the same gas

### 4.1 Joining/disjoining cycles in thermodynamics

Consider a system made of two isolated compartments, each with a volume $V$, filled with the same ideal gas at the same temperature and the same pressure (see Fig.2). Join the two volumes by removing the partition between compartments. This happens without an exchange of work or heat with the surroundings. There are other processes like this, for instance the free expansion of a gas. But in the latter case, it is necessary to provide mechanical work with a piston to restore the system to its initial state. In our case, after joining two volumes of the same gas, it is enough to put the separation back to restore the initial state without any energy expenditure. Thus, the joining of two volumes of identical gas occurs without difference of the Clausius entropy. Note that in the literature, this cycle is most often called mixing/unmixing, even if the gases are identical and there is nothing to mix and unmix, because this refers to another Gibbs paradox that is out of the scope of this paper.

### 4.2 The paradox and its usually proposed solutions

According to the ideal gas law, the number $N$ of particles in each compartment is also the same on average. Initially when compartments are disjoined, since they are isolated the statistical entropy of each consists of the Boltzmann entropy $\ln V^N$. As entropy is additive, the total statistical entropy, $S_0$, of the system is :

$$S_0 = 2N \ln V. \tag{17}$$

Once the two volumes are joined, we have $2N$ particles in a volume $2V$, so that the Boltzmann entropy is $S_1 = \ln(2V)^{2N}$, or :

$$S_1 = 2N \ln V + 2N \ln 2. \tag{18}$$

The difference in statistical entropy is non-zero :

$$\Delta S = S_1 - S_0 = 2N \ln 2. \tag{19}$$

The paradox lies in the apparent contradiction with thermodynamics, as Clausius and statistical entropies are the same, the latter being derived from the former [22].

As an echo to §3.1, note that: 1) the difference of Clausius entropy between the joined and disjoined states is zero; 2) the disjoined state is made of two identical parts. Thus, it can be stated that the entropy of the joined state is twice that of one part of the disjoined state (provided that an absolute values of Clausius entropy makes sense). This result is often viewed as an evidence for the extensivity of Clausius entropy leading to express the above Gibbs paradox in these terms (e.g. [23]). Actually, in thermodynamics the entity in question is $\Delta S$, not $S_0$ or $S_1$ which are meaningless. This expression of the paradox seems to offer an easy solution that does not give us the opportunity to deepen its meaning.

In my knowledge, all solutions to the paradox (see e.g. [12, 15–17, 23–27]) amount in one way or another to write finally instead of Eq. 19:

$$\Delta S = 2N \ln 2 - \ln \binom{2N}{N}. \tag{20}$$

Even if the ways to derive this equation are numerous and correspond to different physical meanings, it can be given a common interpretation in terms of the information (as for the paradox of extensivity) needed to describe the system, or equivalently in terms of the uncertainty about its current microstate (the outcome). In Eq. 20, the term $2N \ln 2$ that originates from the difference in Boltzmann entropy, is due to the increasing number of possible microstates that increases the uncertainty. For one given microstate of the joined system, there are $\binom{2N}{N}$ other possible microstates obtained by the different combinations with respect to the original compartments of particles. At the macroscopic level of thermodynamics, all these microstates are the same. The original compartments of particles is not a relevant information to describe thermodynamically the joined system. So that the variation of the missing information (the increase of uncertainty) has to be reduced by the term $-\ln \binom{2N}{N}$.

The next step of the usual reasoning to solve the Gibbs paradox is to use the Stirling approximation (Eq. 2) that leads to:

$$\begin{aligned} \ln \binom{2N}{N} &= \ln(2N!) - 2\ln(N!) \\ &= 2N \ln(2N/e) - 2N \ln(N/e) \\ &= 2N \ln 2. \end{aligned} \tag{21}$$

Then, with Eq. 20 the statistical entropy of mixing vanishes, consistently with thermodynamics. The paradox is claimed to be solved.

Justifications for Eq. 20 are numerous, basically the same as those to justify the correct Boltzmann counting. Here, the purpose is not to discuss them, but only to point out that the solutions thus proposed are not consistent for the same reason that the usual solutions of the problem of extensivity were not. If the exact Stirling formula (Eq. 4) is used instead of its approximation (Eq. 2), Eq. 21 should be more properly written as:

$$\ln \binom{2N}{N} = 2N \ln 2 - \ln(\sqrt{2\pi N}). \tag{22}$$

So that, by using Eq. 20 the correct result for the difference of entropy should be

$$\Delta S = \ln(\sqrt{2\pi N}), \tag{23}$$

that is non-zero and diverges in the thermodynamic limit. The paradox is actually not solved in this way, but only seems to be because it benefits from a second errors in Eq. 20 and in the way the problem is posed.

## 4.3 Amendment to the solution

In thermodynamics, the variation of Clausius entropy is only defined for reversible transformations (i.e. infinitesimally slow). If a process is not reversible, the corresponding entropy variation can only be determined by the mean of a reversible way to go back to the initial state. Therefore, in the general case, to decide whether a process is reversible or not, it must be considered as belonging to a cycle performed repeatedly and reproducibly. The first cycle of a series cannot be regarded as belonging to such a stationary regime, it must be at least the second for that. In the case of joining/disjoining two volumes of gas, even if initially the numbers of particles in the two compartments are exactly equal, after putting back the separation they differ due to fluctuations and the random repartition of particles between the two compartments. The system is not returned to its initial state. Thus, the first joining/disjoining iteration is not a true cycle, but subsequent iterations are. Therefore, in the framework of statistical mechanics, the joining/disjoining cycle as depicted in Fig.2 is not correct. Figure 3 is the correct representation of what is really happening.

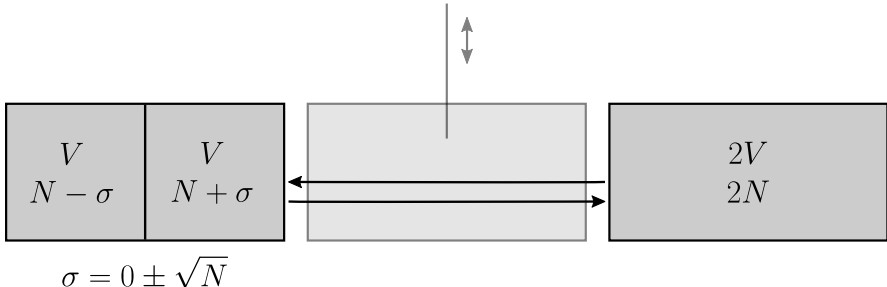

Figure 3: Joining/disjoining cycle as it should be considered in statistical mechanics. The number of particles in each disjoined compartment is known up to $\sqrt{N}$ due to fluctuations, but their sum is constant as the system is isolated.

Fluctuations of the number of particles in each compartment must be taken into account in the calculation of the entropy of the disjoined state. For that, the most direct way is probably the interpretation of entropy in terms of missing information we already used. In describing the disjoined state, a source of uncertainty is the exact number of particles a given compartment has. If this number is known for one, that of the other is also known. It follows that the uncertainty about the disjoined state is in reality increased by a term $\ln(\sqrt{2\pi N})$ (the Shannon entropy of a Gaussian distribution with standard deviation $\sqrt{N}$, i.e. the same as it was in §3.3). There is no equivalent term in the uncertainty about the joined state, since it is closed and its number of particles does not fluctuate. It follows that the joining process decreases the uncertainty by $-\ln(\sqrt{2\pi N})$ and that the difference of entropy of Eq. 20 should be rewritten as :

$$\Delta S = 2N \ln 2 - \ln \binom{2N}{N} - \ln(\sqrt{2\pi N}), \tag{24}$$

that is zero in the thermodynamic limit if we use the exact Stirling formula (Eq. 4 leading to 22). By doing so, the paradox is solved in a consistent manner.

# 5 Conclusion

Considering the exact Stirling formula, instead of its usual approximation, one can see that $\ln(N!)$ has not a linear asymptote. The consequence is that the correct Boltzmann counting alone does not make statistical entropy extensive and does not allow to solve the Gibbs paradox of joining two volumes of the same gas. The correct Boltzmann counting must be accompanied by consideration of fluctuations. In the literature, the linear asymptote is forced by using an approximation of the Stirling's formula, and fluctuations are not considered. So that the solutions proposed for the above paradoxes benefit from the cancellation of these two errors and are logically invalidated.

Despite its obvious successes, statistical mechanics is not, by far, the best-regarded theory in physics. Probably because *"Statistical mechanics is notorious for conceptual problems to which it is difficult to give a convincing answer"* (O. Penrose [28]). The Gibbs paradox, in view of its longevity and the ink it has spilled, can be considered as one of them. In my opinion a rigorous treatment of this paradox, which would start by using the correct asymptotic formula for $\ln(N!)$ and by taking into account fluctuations, would certainly help to be more convincing.

# Acknowledgments

I would like to thank Pierre Lairez who drew my attention to the Stirling approximation *à la physicist* (as he says) and its possible issues.

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
