# Peer review of "Plea for the use of the exact Stirling formula in statistical mechanics"

_SciPost Physics Lecture Notes, doi:SciPost Phys. Lect. Notes 76 (2023)_

## Round 4 · Referee Report · Anonymous (Referee 1) · 2023-8-10

Strengths

The article is structured in a coherent manner, facilitating comprehension.

Weaknesses

While the argument presented is innovative and compelling, the manuscript does exhibit certain shortcomings:
1. As an introductory piece, the depth of content seems misaligned. At some places, rudimentary approximations are presented, whereas in other instances, a more sophisticated understanding is assumed.
2. The scope of the manuscript appears somewhat restricted. If the intention is pedagogical, its utility in guiding students for future research is ambiguous. As it stands, the depth seems inadequate for a lecture note.
3. The presentation seems overly informal for scholarly work. Even when crafting lecture notes, it is essential to maintain a certain standard of formality and writing. Additionally, the manuscript contains several grammatical errors. Recommendations for enhancing clarity are provided in the subsequent section.

Report

In the manuscript entitled " Plea for the use of the exact Stirling formula in statistical mechanics" submitted to SciPost on 17 February 2023, the author offers an insightful resolution to a prominent paradox. The manuscript argues that classical thermodynamics is singularly fixated on equilibrium, overlooking the fluctuations that are introduced by statistical mechanics via probabilities. The work critiques the prevalent usage of the simplified Stirling formula by underscoring its limitations through two principal conflicts: entropy extensivity and Gibbs' paradox pertaining to the amalgamation of identical gases. The manuscript advocates for a reevaluation of our stance on fluctuations in thermodynamics and the precision of the Stirling formula for a more accurate understanding.

In this scholarly piece, the initial discourse revolves around the introduction of the Stirling formula and its approximated version. Subsequently, there is an elucidation of the concept of extensivity in both thermodynamics and statistical physics. Finally, the manuscript addresses two classical paradoxes using the precise Stirling formula, complemented by an interpretation through information theory.

To further elucidate the listed weaknesses, the following inconsistencies were identified:
- At equation (2), (3), stating the Riemann sum here would help students and evaluating them can be left as exercises. “… integral … between Riemann sums, or equivalently … sum … squeezed between two integrals” does not provide clarity.
- After equation (4), the sub-leading term is identified with the Shannon entropy in a Gaussian distribution. Is it assumed that readers are familiar with information theory, but not quantum mechanics or statistical physics? This is a central argument in the paper; however, this is taken as a fact without any reference or explanation.
- After equation (4), why is Wallis integral relevant here? It's worth noting that physicists might be more acquainted with the Gamma function than with these particular integrals.
- Derivation from (5) to (10) seems to long just to explain extensivity. If readers are assumed to have a grasp on both thermodynamics and statistical physics, such a prolonged exposition appears redundant.
- In equation (11), it is understandable that the author intends to use a very simple formula for entropy. However, this is not the Boltzmann entropy. Furthermore, the rationale behind the formula uses de Broglie concept, which is quantum in nature.

Given the two acceptance criteria:
- Cover a subject of ongoing interest to the research community.
- Provide a correct, systematic, and intelligible presentation of the material.
As the depth of the paper is lacking, it does not meet the first criterion. The second criterion would be met if the author addresses the informal writing style as well as grammatical errors. Overall, the reviewer does not recommend this manuscript for publication under SciPost Physics Lecture Notes. This article might be better suited for AJP - AAPT.

Requested changes

The succeeding section provides recommendations to rectify typographical errors and enhance the clarity of the manuscript (note that numbering resets at section beginnings, subsequent to equations, or at page beginnings).
Intro passage:
Position: Text - Suggestion
Line 5-10 “... the value of entropy at equilibrium cannot be compared to those close to equilibrium …” - Expand on what “cannot be compared to” means.
Line 12 “... it is quite amazing that a rough approximation … is commonly used …” - Replace “amazing” by “intriguing”, “paradoxical”, etc.
Line 22 “...one of Gibbs’ well-known paradoxes, that concerning the joining ...” - “...one of Gibbs’ well-known paradoxes that concerns the joining ...”
Line 25 “…considering them independently each other …” - “…considering them independently ...”
Line 26 “We will see that the usual treatments of these paradoxes, benefit from …” - Drop the comma
Line 27 “1) that of neglecting … 2) that of using …” - Drop the two “that of”
Line 31 “…are not resolved like this, but can be …” - Clarify what “like this” mean

Section I.
Position Text Suggestion
Line 3 from (3) “… is not convergente, contrary to …” - Typo
Line 4 from(3) “(see [2] p.53).” - Use more recent edition.
Line 5 from (3) “It is the calculation of the limit of the latter that is actually due to Stirling who proved that …” - “The actual calculation of the latter’s limit is attributed to Stirling, who proved that …”. Citation is recommended here.

Section II.A
Position Text Suggestion
Line 1 from (7) “If u and v are constant, then s is too …” - “If u and v are constants, then so is s, …”
Line 2, right column, page 2 “… write the fundamental equality dS = …” - “… write the fundamental equality as dS = …”
Line 6-7, … “It is the contribution of statistical mechanics to have made …” - “Statistical mechanics has made …”
Line 13-14, … “But there is no more evidence to the contrary.” - “Nor is there evidence to the contrary.”
Line 26, … “(specially statistical mechanics)” - “(especially statistical mechanics)”
Line 32 & 38, … “… the entropy … is maximum at equilibrium” - “… the entropy … is maximized at equilibrium”
Line 36-37, … “(traditionally the second law of thermodynamics)” - Include a comma after. Recommend breaking this sentence further.
Line 42, … “… maximizing the entropy of a system supposes that …” - “… maximizing the entropy of a system assumes that …”
Line 46, … “They are necessarily in equilibrium each other ...” - “… with each other …”
Line 48-49, … “So that the entropy … can be computed whatever …” - Change “whatever” to “regardless of the values of”

Section II.B
Position Text Suggestion
Line 3 “consistently with …” “consistent with …”
Line 6 from (11) “Before to see how …” “Before examining how…”
Line 18 from (11) “… all solutions to solve the …” “… all solutions to the … “
Line 8, right column, page 3 “… Boltzamann counting …” “… Boltzmann counting …”
Line 10, right column, page 3 “… Stirling’s approximation like this : . …” Drop the colon.

Section II.C
Position Text Suggestion
Line 6-7 “… not easily conceivable in his world.”- “… not easily conceivable in his time.”
Line 16 “As everything … have been built …” - “As everything … has been built …”
Line 17 “… for reason of self-consistency …” - “… for self-consistency …”
Line 20 “… to follows their trajectories, …” - “… to follow their trajectories, …”
Line 26 “Thus any …” - “Thus, any …”
Line 29 “A constant that will vanish …” - “This constant will vanish …”
Line 39 “… distinguishable particles … gives …” - “… distinguishable particles … give …”
Line 40 “… undistinguisable … “ - “… indistinguishable …”
Line 13, left column, page 4 “Let us think the state …” - “Let us consider the state …”
Line 15, … “… [17?, 18]…” - Fix the reference
Line 17, … “… and the outcome that one the system is currently adopting.” - Drop “one”
Line 22, … “… one given microstate …” - “… a given microstate …”
Line 25, … “… which exact microstate …” - “… which specific microstate …”
Line 28, … “… must by subtracted …” - “… must be subtracted …”
Line 33, … “This number obeys to a …” - “This number obeys a …”

Section III.A
Position Text Suggestion
Line 2 “… of same volume V …” - “…, each with a volume V, …”
Line 3 “… removing the separation …” - “… removing the partition …”
Line 10 “… of the same gas ...” - “… of identical gas …”

Section III.B
Position Text Suggestion
Line 4 after (19) “1) as the … 2) as the …” - Remove the “as the”s
Line 2, left column, page 5 “So that this result …” - “This result … ”
Line 6, left column, page 5 “So that the paradox, expressed like this, finds a solution …” - “This expression of the paradox seems to offer an easy solution …”
Right before (20) “…all solutions proposed to solve ...” - “… all solutions to …”

Section III.C
Position Text Suggestion
Line 2 “So that in the general case …” - “In general, …”
After (24) “Doing so the paradox is solved …” - “By doing so, the paradox is solved …”

  • validity: ok
  • significance: low
  • originality: good
  • clarity: low
  • formatting: reasonable
  • grammar: below threshold

Author:  Didier Lairez  on 2023-08-10  [id 3895]

(in reply to Report 1 on 2023-08-10)

First of all, I want to thank you very much for your careful reading and for all your constructive suggestions to improve the style and grammar of the paper. All requested changes are of course taken into account in the revised version.

As for the major criticism contained in your report, it concerns the target reader of the paper.
Indeed, it is assumed that he has basic knowledge of what entropy is. Thus, that he has basic knowledge about thermodynamics, statistical mechanics and information theory (i.e. all fields where entropy is introduced). This last one is probably the least known by physicists. So that I agree that a reference about the Shannon's entropy of a Gaussian distribution was missing. This is now corrected.

If I understand your criticism correctly, physicists should be so aware about thermodynamics and statistical mechanics, so that section II-A seems too long, a bit boring, and ultimately suggests that the target reader is a student. In my opinion, this is not the case. After discussion with many colleagues, I realized that there was a frequent confusion between "additivity" and "extensivity". This confusion can even be found in the famous (and very good) book of Callen. The distinction between these two properties is a key point to understand why extensivity has been taken as a postulate in axiomatic thermodynamics.
I believe that the "pedestrian" side of section II-A is far from useless.

Finally, I do not understand what exactly is your judgement on the scientific content of the paper. In one side, you acknowledge its "validity" and "originality" (the latter being not so easy to obtain with so old subjects), so that in terms of the information it contains one may expect that its "significance" should be great. But on the other side you state that its "significance" is low...

Anonymous on 2023-08-10  [id 3896]

(in reply to Didier Lairez on 2023-08-10 [id 3895])
Category:
answer to question

To address the author's comments:

"If I understand your criticism correctly, physicists should be so aware about thermodynamics and statistical mechanics, so that section II-A seems too long, a bit boring, and ultimately suggests that the target reader is a student. In my opinion, this is not the case. After discussion with many colleagues, I realized that there was a frequent confusion between "additivity" and "extensivity". This confusion can even be found in the famous (and very good) book of Callen. The distinction between these two properties is a key point to understand why extensivity has been taken as a postulate in axiomatic thermodynamics.
I believe that the "pedestrian" side of section II-A is far from useless."

As "usefulness" is a subjective concept, the reviewer retract their comment on this section. While some may find an in-depth discussion redundant, it could prove invaluable to others.

"Finally, I do not understand what exactly is your judgement on the scientific content of the paper. In one side, you acknowledge its "validity" and "originality" (the latter being not so easy to obtain with so old subjects), so that in terms of the information it contains one may expect that its "significance" should be great. But on the other side you state that its "significance" is low..."

To clarify, while we think the manuscript is delightful to read, its relevance and "significance" to our platform seems misaligned. Our objective is to feature content that not only addresses particular questions in classical subjects but also offers substantial resources for both novices and experts venturing into contemporary research fields. Consequently, I suggest considering platforms like the AAPT, or similar outlets, as a more appropriate venue for this manuscript.

Anonymous on 2023-08-11  [id 3899]

(in reply to Anonymous Comment on 2023-08-10 [id 3896])

You say that the scope of the article is not broad enough ("The scope of the manuscript appears somewhat restricted", [it does not] "Cover a subject of ongoing interest to the research community", [it does not] "offers substantial resources").

It is true that the article is more interested in the root of the theory than in its ultimate leaves. For this reason, although the length of the article is "restricted", its topic is not:

1) the article solves an error that spreads over time in all thermostatistics textbooks up to the most recent;

2) [the article] "offers an insightful resolution to a prominent paradox" (sic). Let me add "prominent" and very old, in fact present in the theory from the beginning.

Your proposal to submit in a journal like AJP-AAPT is probably a good idea in a perfect world. But from a pragmatic point of view (I am sorry to have obligations towards my employer), the paper has been submitted to SciPost 6 months ago and your report is the first I have received. This demonstrates the difficulty of finding reviewers in this field. I am afraid that restarting the procedure from zero will not help...

Finally, this demonstrates also the difficulty of publishing on subjects outside "contemporary research fields". "Contemporary research" is like "modern research", it does not stay modern for long. So in some ways it is also "somewhat restricted".

---

## Editorial Decision

published